# CLEVER: A Curated Benchmark for Formally Verified Code Generation

**Amitayush Thakur**[†]
amitayush@utexas.edu

**Jasper Lee**[†]
leejasper@utexas.edu

**George Tsoukalas**[†]
george.tsoukalas@utexas.edu

**Meghana Sistla**[†]
mesistla@utexas.edu

**Matthew Zhao**[†]
matthewzhao@utexas.edu

**Stefan Zetzsche**[‡]
stefanze@amazon.co.uk

**Greg Durrett**[†]
gdurrett@cs.utexas.edu

**Yisong Yue**[⋆]
yyue@caltech.edu

**Swarat Chaudhuri**[†]
swarat@cs.utexas.edu

[†] UT Austin      [‡] Amazon      [⋆] Caltech

## Abstract

We introduce CLEVER[1], a high-quality, curated benchmark of 161 problems for end-to-end verified code generation in Lean. Each problem consists of (1) the task of generating a specification that matches a held-out ground-truth specification, and (2) the task of generating a Lean implementation that provably satisfies this specification. Unlike prior benchmarks, CLEVER avoids test-case supervision, LLM-generated annotations, and specifications that leak implementation logic or allow vacuous solutions. All outputs are verified post-hoc using Lean's type checker to ensure machine-checkable correctness. We use CLEVER to evaluate several few-shot and agentic approaches based on state-of-the-art language models. These methods all struggle to achieve full verification, establishing it as a challenging frontier benchmark for program synthesis and formal reasoning. Our benchmark can be found on GitHub as well as HuggingFace. All our evaluation code is also available online.

## 1 Introduction

Interactive theorem-provers (ITPs) [11, 29, 6] are an established technology for engineering high-assurance software, leading to success stories like the CompCert verified C compiler [21] and the seL4 [16] verified microkernel. However, writing formal specifications and correctness proofs for software systems can take tremendous effort — for example, the development of seL4 was reported to take 20+ person-years. These costs are a key impediment to the broad deployment of ITP-based formal verification.

Recent progress in autoformalization and neural theorem-proving [30, 22] has raised hopes of scaling up formal verification [37]. Most existing work in this area has focused on formalizing and proving statements in pure mathematics [38, 32]. However, the software verification setting opens

---

[1]CLEVER: Curated Lean Verified Code Generation Benchmark

39th Conference on Neural Information Processing Systems (NeurIPS 2025) Track on Datasets and Benchmarks.

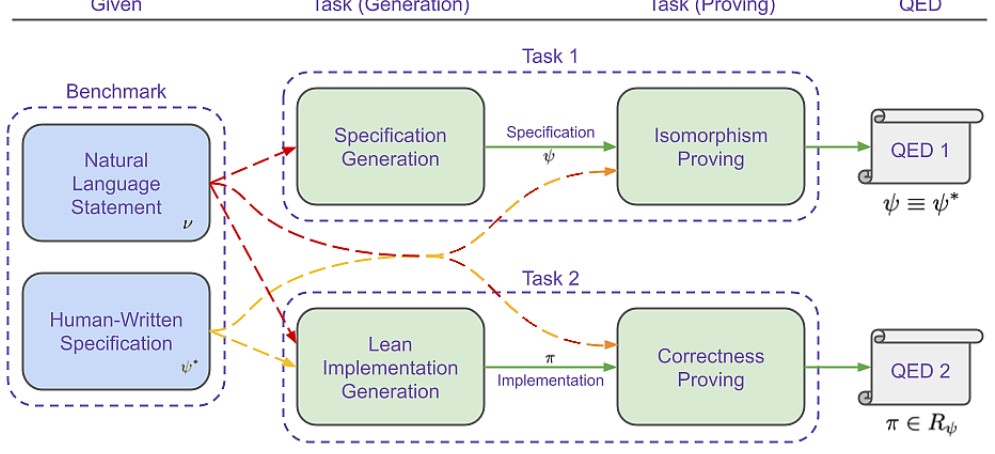

Figure 1: The two tasks of the CLEVER benchmark pipeline. Task 1 requires first generating a specification $\psi$ from the natural language statement $\nu$, then proving an isomorphism between the generated specification and a human-written specification $\psi^*$. Task 2 requires first generating a Lean implementation $\pi$, then proving its correctness according to the human-written specification. Both of these tasks must be completed correctly (reaching both QED 1 and QED 2) in order for a success to be counted.

up the challenge of *generating code that is formally verified by construction*, a problem without a well-studied analog in the mathematics setting.

To date, there are a handful of benchmarks [9, 27, 26] for formally verified code generation. However, the formal specifications in these benchmarks tend not to capture the full (natural-language) intent behind the target program and sometimes hint at ways to implement the program. This ambiguity allows a code generator to "cheat" by generating trivial programs or copying code from the specification (see Appendix A.1).

In this paper, we address this gap in the prior art with CLEVER, a high-quality benchmark for formally verified AI-based code generation. CLEVER includes hand-crafted Lean specifications of 161 programming tasks from the HUMANEVAL benchmark [4].

It evaluates models in two stages: (1) *Specification certification:* Given a natural language specification, the model is required to generate a Lean specification and prove that it is semantically equivalent to the ground-truth specification. (2) *Implementation certification:* Once the model has correctly generated the specification, it is required to generate a Lean implementation and prove that it satisfies the ground-truth specification. A synthesis attempt is deemed *successful* only when both the proofs generated in the two stages are fully verified by Lean's type checker. This rigorous pipeline avoids the pitfalls of both automatically generated specifications and test-based supervision.

We use CLEVER to evaluate several state-of-the-art LLMs prompted in a few-shot manner and show that they can only solve up to 1/161 end-to-end verified code generation problem, establishing CLEVER as a challenging frontier benchmark for program synthesis and formal reasoning. In summary, our contributions include:

1. We introduce CLEVER, the first curated benchmark for evaluating the generation of specifications and formally verified code in Lean. The benchmark comprises of 161 programming problems; it evaluates both *formal specification generation* and *implementation synthesis* from natural language, requiring formal correctness proofs for both. All specifications are manually written to be complete, implementation-agnostic, and free from exploitable artifacts, preventing models from shortcutting the intended semantics.

2. We present an empirical evaluation of several state-of-the-art LLMs and agentic approaches on CLEVER and show that they all struggle at meeting the benchmark's goals, establishing the challenging nature of the benchmark.

```
(a)                                              (b)

-- computable spec                               -- non-computable spec
def problem_spec                                 inductive fibonacci_non_computable : ℕ → ℕ → Prop
-- function signature                            | base0 : fibonacci_non_computable 0 0
(implementation: Nat → Nat)                      | base1 : fibonacci_non_computable 1 1
-- inputs                                        | step  : ∀ n f₁ f₂,
(n: Nat) : Prop :=                               fibonacci_non_computable n f₁ →
-- spec                                          fibonacci_non_computable (n + 1) f₂ →
let spec (result: Nat) :=                        fibonacci_non_computable (n + 2) (f₁ + f₂)
match n with
| 0 => result = 0                                def problem_spec
| 1 => result = 1                                -- function signature
| n' + 2 => result =                             (implementation: Nat → Nat)
implementation n' +                              -- inputs
implementation (n' + 1)                          (n: Nat) :=
-- return value satisfies spec                   -- spec
∃ result, implementation n = result ∧ spec result    let spec (result: Nat) :=
                                                   fibonacci_non_computable n result
                                                 -- program termination
                                                 ∃ result,
                                                   implementation xs = result ∧ spec result
```

Figure 2: Two different specs for finding the $n^{th}$ Fibonacci number. (a) shows a computable specification that *leaks* the implementation; (b) shows a non-computable specification leading to no-leakage of the implementation and enforcing the model to learn the deeper logical inference.

## 2   The CLEVER Benchmark

CLEVER builds on HUMANEVAL [4] by adapting 161[2] of its 164 programming problems for formal verification in Lean 4. Each problem includes a natural language description ($\nu$), a human-authored formal specification ($\psi^*$), a Lean function signature ($\pi_{\text{sig}}$) for the implementation, and Lean theorems for both specification equivalence and implementation correctness. All formal specifications are written as *non-computable* logical propositions — i.e., they use quantifiers and logical connectives that cannot be directly evaluated — ensuring that models cannot copy implementation logic from specification syntax.

During evaluation, a model being evaluated on the benchmark starts with the natural-language description $\nu$. Given this text, the model must generate:

(1) a formal Lean specification $\psi$, expressed as a predicate (a function that returns a Lean 4 proposition i.e. Prop),

(2) a proof that $\psi$ is semantically equivalent to a hidden ground-truth Lean specification $\psi^*$,

(3) a Lean implementation[3] $\pi$ that matches the function signature ($\pi_{\text{sig}}$) and is designed to satisfy $\psi^*$ (and hence $\psi$), and

(4) a formal proof establishing that $\pi$ satisfies $\psi^*$.

These steps (Figure 1) form two certification goals: (1) *Specification certification:* Steps 1–2 verify that the model correctly inferred the intended behavior. (2) *Implementation certification:* Steps 3–4 verify that the generated implementation satisfies the formal intent.

Our staged reasoning setup allows fine-grained diagnosis: models may fail at generating specifications, proving equivalence between the generated and ground-truth specifications, synthesizing implementations, or proving implementation correctness. For example, note that we require the generated implementation $\pi$ to satisfy the ground-truth specification $\psi^*$ instead of the model-generated specification $\psi$. This is because we want the evaluation of $\pi$ to be independent of the ability of the model to generate the correct specification. More generally, failures at the various stages of our pipeline are independently diagnosed using Lean's type checker.

---

[2]Not all problems could be formalized due to limitations in Lean 4 and its supported libraries.

[3]Here, we use the fact that Lean is not just a language for mathematical specifications and proofs but a full-fledged functional programming language.

```
(a)

def problem_spec
(implementation: List Int → Int → Bool)
(q: List Int) (w: Int) :=
let spec (result : Bool) :=
  result ↔ (List.Palindrome q) ∧ (List.sum q ≤ w)
∃ result, implementation q w = result ∧ spec result

(b)

def implementation (q: List Int) (w: Int) : Bool :=
-- implementation generated by GPT-4o
List.Palindrome q ∧ List.sum q ≤ w

(c)

theorem correctness (q: List Int) (w: Int)
: problem_spec implementation q w := by
-- proof generated by GPT-4o
unfold problem_spec
let result := implementation q w
use result
simp [result]
simp [implementation]
```

```
(d)

def problem_spec
(implementation: List Int → Int → Bool)
(q: List Int) (w: Int) :=
let spec (result : Bool) :=
  (result → (List.Palindrome q)) ∧
  (result → (List.sum q ≤ w)) ∧
  (¬(List.Palindrome q) → ¬ result) ∧
  (¬(List.sum q ≤ w) → ¬ result)
∃ result, implementation q w = result ∧ spec result

(e)

def implementation (q: List Int) (w: Int) : Bool :=
-- implementation generated by GPT-4o
let is_palindrome := q = q.reverse
let sum_le_w := q.sum ≤ w
is_palindrome && sum_le_w

(f)

theorem correctness
(q: List Int) (w: Int)
: problem_spec implementation q w
:= by
-- proof generated by GPT-4o
unfold problem_spec
let result := implementation q w
use result
simp [result]
simp [implementation]
intro h -- <- The compilation fails here
simp [h]
exact List.eq_reverse_of_palindrome h.left
-- more proof trimmed
```

Figure 3: Illustration of specification leakage (left) and its mitigation (right) via non-computable specifications, using HUMANEVAL problem 72. The task is to return true iff a list q is a palindrome and its sum is at most w. In (a–c), the spec is *computable*: it encodes the desired logic in a Boolean expression, allowing the model to copy it directly in (b) and produce a trivial proof (c) via just unfolding and simplifying basic definitions used in the theorem statement. In contrast, (d–f) use a *non-computable* spec expressed in Prop with logical implications. The corresponding implementation (e), generated by GPT-4o using few-shot prompting, reflects the semantic intent without mirroring the spec. The proof (f) fails without additional reasoning, highlighting the challenge of proving correctness when logic cannot be mechanically unfolded. Non-computable specs thus act as guardrails, requiring models to reason rather than copy.

**Challenges Encountered during Formalization.** A key design decision in our benchmark is the use of *non-computable* specifications, which are predicates or functions in Lean that return propositions (Prop in Lean) that cannot be evaluated or simplified (decided by Lean) through computation alone. These contrast with *computable* specifications, written as executable functions or decidable predicates that Lean can reduce directly. While easier to verify, computable specs often *leak* the desired logic: models can copy them into implementations and produce trivial proofs via rewriting. Figure 2 shows the difference between a computable and a non-computable specification.

Figure 3 demonstrates the importance of this contrast. The left side (a–c) shows a computable spec whose logic is mirrored exactly in the GPT-4o-generated implementation, enabling a trivial proof. On the right (d–f), the spec is non-computable and requires symbolic reasoning to prove correctness. Notably, the GPT-4o-generated implementation in (e) does not mirror the spec, and the proof fails without further reasoning. This design ensures that models must engage in deeper logical inference, not just syntactic pattern matching. By using non-computable specs across our benchmark, we eliminate leakage and enforce truly verified reasoning from models.

Creating this benchmark involved substantial manual effort. On average, writing a formal specification took annotators *25 minutes per problem* on average, with an additional 15 minutes spent reviewing each other's specifications. Some problems involving complex non-computable specs required over an hour. To better understand problem difficulty and verify feasibility, we manually authored correctness proofs for a small random sample of benchmark problems. These ranged from *10 lines (e.g., problem_17) to 225 lines (e.g., problem_0)*, reflecting a wide span of proof complexity.

```
(a)                                              (b)
def problem_spec                                 -- possible implementation using Newton's method
-- function signature                            def implementation (xs: List Rat) : Rat :=
(implementation: List Rat → Rat)                 let rec poly (xs: List Rat) (x: Rat) := xs.reverse.
-- inputs                                            foldl (λ acc a => acc * x + a) 0;
(xs: List Rat) :=                                let rec poly' (xs: List Rat) (x: Rat) := (xs.drop 1).
-- spec                                              reverse.foldl (λ acc a => acc * x + a) 0;
let spec (result: Rat) :=                        let rec eps := (1: Rat) / 1000000;
  let eps := (1: Rat) / 1000000;                 let rec find_zero (xs: List Rat) (guess: Rat) (fuel:
  xs.length ≥ 1 → xs.length % 2 = 0 →                 Nat) :=
  ∀ poly : Polynomial Rat,                       let eval := poly xs guess;
    poly.degree = some (xs.length - 1) →         let eval' := poly' xs guess;
    (∀ i, i ≤ xs.length - 1 → poly.coeff i = xs.get! if eval ≤ eps ∨ fuel = 0 then (guess, fuel)
      i) →                                       else
    |poly.eval result| ≤ eps;                    let guess' := (eval' * guess - eval) / eval';
-- program termination                           find_zero xs guess' (fuel - 1);
∃ result,                                        (find_zero xs 1.0 1000000).1
  implementation xs = result ∧
  spec result
```

Figure 4: **Polynomial Root-Finding.** Problem 32 asks for an approximate real root of a degree-$n$ polynomial. The spec enforces small residual error ($< 10^{-6}$). The implementation uses Newton's method with bounded recursion; proving termination is non-trivial due to lack of guaranteed derivative behavior.

In addition to the main benchmark, we *release a small hand-curated few-shot prompt dataset* comprising of 5 problems distinct from HUMANEVAL. All of these problems include hand-written implementations, and some of them additionally include manually written equivalence and isomorphism proofs. For example, one correctness proof spans 309 lines, while corresponding isomorphism proofs range from 29 to 82 lines. This auxiliary dataset is intended to support prompt tuning and evaluation in few-shot or in-context learning setups.

Curating the benchmark also revealed deeper challenges inherent to formal verification. For instance, in the HUMANEVAL problem involving root-finding for polynomials (see Figure 4), proving termination is difficult due to reliance on unbounded numerical search. Similarly, generating verified code for "finding all prime Fibonacci numbers" encounters foundational roadblocks, as there is no known proof that infinitely many such numbers exist—highlighting how natural language tasks can conceal deep mathematical issues when formalized. One potential way to deal with these types of formulations is by adding the concept of computational fuel and approximate answers (see Figure 4, and Figure 9 in Appendix A.2). Writing *non-computable* specifications is particularly challenging for problems that rely on language-level features like Python's eval, as seen in Problem 160. Since Lean lacks direct string-based evaluation, we had to reconstruct the behavior using inductive definitions over token lists and arithmetic expressions. This required converting a naturally computable task into a semantically equivalent, non-computable formulation without leaking implementation details. As shown in Figure 11 (in Appendix A.3), achieving this often involves layered recursive structures and careful abstraction to ensure both correctness and opacity.

Another instructive case is the problem of computing the MD5 checksum (problem 162). Here, the formal specification must, by necessity, describe the exact computation, making it closely related to the implementation itself. Since we could not find any popular hashing libraries in Lean, we chose not to formalize this specific problem. However, we prescribe the recipe for creating non-computable definitions in Appendix A.3, given that we know the computable definition.

While adapting HUMANEVAL to Lean, we encountered several language-level limitations. Some problems relying on dynamic typing or polymorphic return types—like Python's Any—could not be faithfully represented in a statically typed setting (e.g., problems 22 and 137). As a result, we were able to formalize 161 out of the original 164 problems. In problem 103, where the output is either a binary string or None based on input validity, we use Option String as the return type. In problem 129, where the function may return either a list of words or a number, we encode this using disjoint union type in Lean: (List String) ⊕ Nat, allowing only one of the two values to be populated at a time.

Prior work, such as FVAPPS [9], relies on automatically generated specifications that can be incomplete or leaky, allowing trivial implementations (e.g., always returning zero) to pass (see Figure 8 in Appendix A.1). Our human-curated specifications ensure completeness and robustness, closing such loopholes and surfacing the real verification complexity hidden in everyday programming problems.

| | Approach Components | | | | Pass@k-sec Spec Cert. | | Impl Cert. | | End-to-End |
|---|---|---|---|---|---|---|---|---|---|
| Model | Spec Gen | Equiv Proof | Impl Gen | Corr Proof | *Compiled* | *Proved* | *Compiled* | *Proved* | |
| | | *Few-Shot Baseline* | | | | | | | |
| GPT-4o | FS | FS | FS | FS | 84.472% | 0.621% | 68.323% | 0.621% | 0% |
| o4-mini | FS | FS | FS | FS | 82.609% | 1.242% | 83.230% | 1.863% | 0.621% |
| Claude-3.7 | FS | FS | FS | FS | 86.957% | 0.621% | 65.217% | 1.863% | 0.621% |
| DeepSeek-R1 | FS | FS | FS | FS | 71.42% | 0.621% | 60.870% | 5.559% | 0.621% |
| | | *COPRA Baseline* | | | | | | | |
| GPT-4o | FS | COPRA | FS | COPRA | 76.398% | 1.863% | 68.323% | 3.727% | 0.621% |
| Claude-3.7 | FS | COPRA | FS | COPRA | 81.366% | 1.242% | 65.217% | 8.696% | 0.621% |
| | | *Hybrid Baseline* | | | | | | | |
| GPT-5-mini | FS | Kimina FS | FS | Kimina FS | 90.062% | 0% | 84.472% | 0.621% | 0% |

Table 1: Evaluation of different strategies for **end-to-end verified code generation**. Each approach consists of five components: **Model** (LLM used), **Spec Gen** (formal specification generation), **Equiv Proof** (proof of equivalence to ground-truth spec), **Impl Gen** (program synthesis), and **Corr Proof** (proof of implementation correctness). **FS** indicates few-shot prompting with 1–2 examples. Evaluation follows the pipeline in Figure 5. **Pass@k-seconds** with $k = 600$ reports the fraction of tasks where Lean successfully compiles the outputs and accepts the associated proofs within a 600-second time budget. The **Compiled** columns indicate whether the generated Lean code is syntactically valid and type-checks. The **Proved** columns reflect whether the corresponding proofs were accepted by Lean's kernel, thereby certifying semantic correctness. The **End-to-End** column reports full pipeline success—i.e., both the specification and implementation must compile and their respective proofs must be accepted. Despite strong models like GPT-4o achieving high compilation rates, formal correctness remains challenging: no approach has yet succeeded across all stages on more than one problem (specifically problem 53).

## 3 Evaluation

We evaluated several state-of-the-art LLMs and agentic approaches on CLEVER. Now we elaborate on the results.

**Evaluation Metric.** To fairly compare approaches that differ in model size, latency, and API usage, we adopt the metric *pass@k-seconds*—the fraction of benchmark problems solved within a fixed time budget $k$. A task is marked as solved only if both the formal specification and the implementation are generated and verified via Lean's type checker. As described in Figure 5, each step in the CLEVER pipeline (spec generation, equivalence proof, implementation, and correctness proof) is retried until a valid Lean-compilable output is found or the time runs out.

We also compare the different approaches for each task using *pass@k*[4] (*compile@k* and *prove@k* respectively).

**Evaluated Baselines.** We evaluate three families of approaches for end-to-end verified code generation. The **Few-Shot Baseline** uses large language models (GPT-4o, Claude-3.7, o4-mini, and DeepSeek-R1) to generate all components—specifications, implementations, and proofs—via few-shot prompting with 1–2 exemplars. This baseline assesses the raw capability of LLMs to reason formally without task-specific training or tooling. The **COPRA Baseline** replaces the proof generation steps (Stages 2 and 4) with COPRA [31], a neuro-symbolic proof search agent designed to produce Lean-compatible proofs when provided with an off-the-shelf foundational model and a Lean theorem statement to prove. This setup isolates proof search difficulty from the upstream generation task.

EVALUATE($approach$, timeout)

```
 1  ▷ Assume RETRY retries the given function
 2  ▷ until it generates compilable Lean 4 code or timeouts.
 3  ▷ RETRY returns the Lean 4 code and remaining time.
 4  t_rem ← timeout
 5  ψ, t_rem ← RETRY(GenerateSpec, ν, t_rem)
 6  P_eq, t_rem ← RETRY(ProveEquivalence, (ψ, ψ*), t_rem)
 7  if t_rem ≤ 0 return Fail
 8  π, t_rem ← RETRY(GenerateImpl, (ν, ψ), t_rem)
 9  P_χ, t_rem ← RETRY(ProveCorrectness, (π, ψ*), t_rem)
10  if t_rem ≤ 0 return Fail
11  return Success (all Lean 4 checks passed)
```

Figure 5: Evaluation strategy: retry each generation step until Lean compilation succeeds or a timeout is reached.

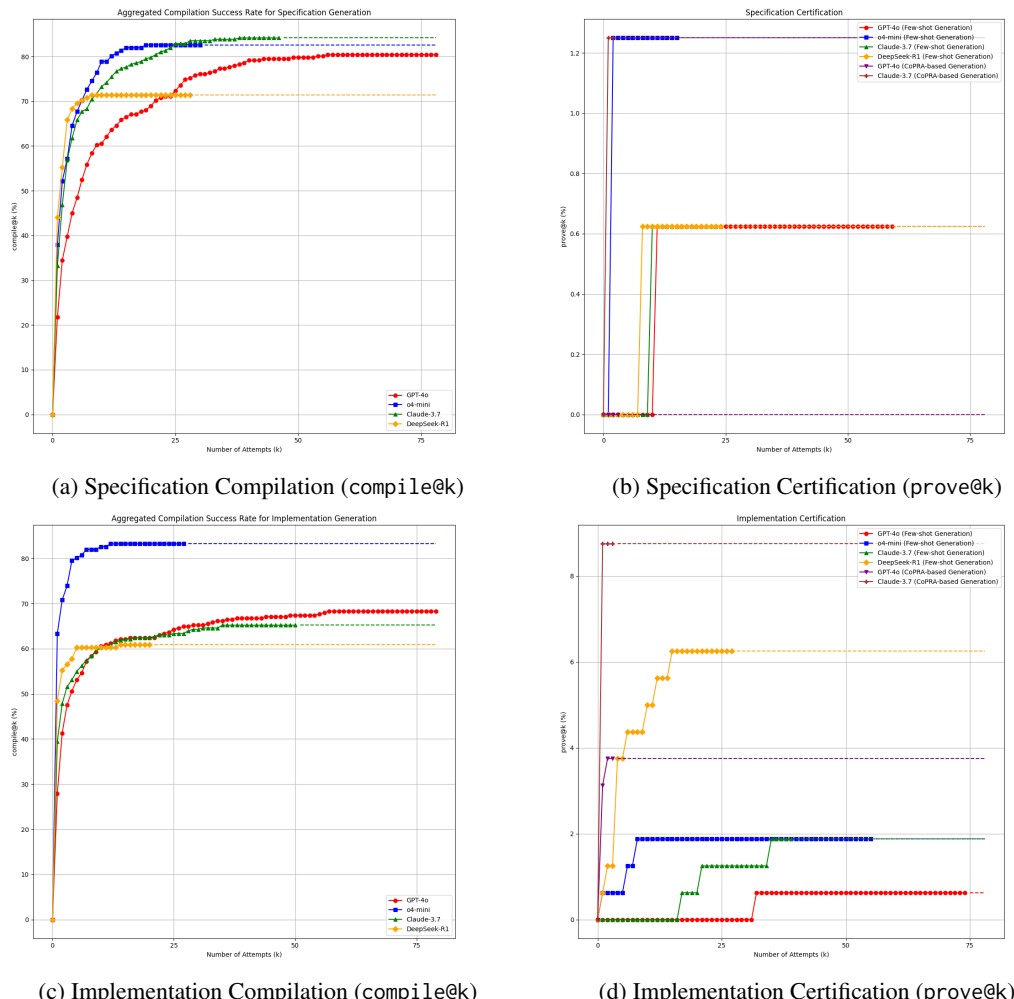

(a) Specification Compilation (`compile@k`)

(b) Specification Certification (`prove@k`)

(c) Implementation Compilation (`compile@k`)

(d) Implementation Certification (`prove@k`)

Figure 6: Aggregated `compile@k` and `prove@k` results across $k$ attempts, diagnosing failure modes for specification and implementation generation. Dotted lines indicate extrapolation of the pass rate beyond the 600s timeout. **(a) Specification Compilation:** Most models achieve high `compile@k` rates, with `Claude-3.7` and `o4-mini` reaching >80%. **(b) Specification Certification:** Proving specification equivalence is a major bottleneck. All few-shot models solve only one problem (0.62%), while `GPT-4o` (CoPRA) proves 1.8%. **(c) Implementation Compilation:** `o4-mini` has the highest compilation rate at >80%, while other models cluster between 60-70%. **(d) Implementation Certification:** `Claude-3.7` (CoPRA) performs best, certifying 8.7% of implementations, followed by `DeepSeek-R1` (Few-shot) at 5.6%. These plots show that while models are fluent at generating *compilable* artifacts, formal *certification* remains the key challenge, with all `prove@k` rates below 10%.

**Results.** Our primary evaluation metric focuses strictly on semantic correctness: a task is considered successful only if both the specification and the implementation are formally certified via Lean proofs. This strict definition ensures that reported scores reflect genuine end-to-end verification. However, to better diagnose failure modes, we also report auxiliary statistics: the fraction of tasks where generated specifications and implementations *compile* successfully. These serve as proxies for the model's fluency in Lean and its ability to produce well-typed artifacts.

In particular, implementation compilation includes not only type-checking against the declared function signature, but also validation against a suite of example-based test cases adapted from the original HUMANEVAL prompts. While passing these tests provides some evidence of functional correctness[25], we deliberately exclude them from our core success metric—since test cases offer only partial coverage and cannot guarantee semantic soundness (see Section 2 for discussion).

As shown in Figure 6 and Table 1, compilation rates are broadly similar across few-shot models for both specification (Figure 6a) and implementation (Figure 6c) generation. A notable exception is the higher implementation compilation rate achieved by o4-mini, which contrasts with its lower success in proving correctness. More generally, even when an approach successfully certifies multiple specifications (Figure 6b) or verifies correctness for multiple implementations (Figure 6d), the overall end-to-end success rate remains low. This is largely due to *mismatch*: tasks for which specification certification is tractable are often those where implementation correctness proofs are especially difficult, and vice versa. As a result, the joint success condition is rarely satisfied.

Another interesting observation is that Claude-3.7, when used along with COPRA, can certify more implementations (14) than all other models; however, its performance on specification certification is only comparable to other models. We believe that this might have to do with the length of proofs needed for specification certification, and hence, in the limited timeout it is hard to find the full proof for specification.

Finally, we evaluated *KiminaProver-7b*[33], a specialized prover. Due to its non-standard output formatting which posed parsing challenges, we used it only for proof steps, pairing it with GPT-5-mini for generation. This hybrid (Table 1) yielded the **highest compilation rates** (90.1% spec, 84.5% impl), highlighting GPT-5-mini's fluency. However, it certified almost nothing (0 spec, 1 impl), as the pipeline was bottlenecked by these same parsing issues.

**Proof Difficulty and Structure.** As shown in Table 2, proofs for **specification certification** are consistently longer and harder to generate than those for implementation correctness. This is expected: proving that a generated spec is semantically equivalent to a non-computable reference specification requires models (or agents) to reason abstractly about intent, without access to implementation-level cues. In contrast, correctness proofs often benefit from direct pattern matching or automation through tactics like simp.

This distinction is especially evident in the only problem for which an end-to-end verified code generation succeeds across multiple models: **problem 53**, which asks for the sum of two integers. Despite the simplicity of the implementation, the ground-truth specification is expressed in a way that deliberately obfuscates the target behavior. This design makes the equivalence proof non-trivial and requires models (or COPRA) to recover the algebraic structure underlying addition. Even here, success is only possible because the proofs admit aggressive automation via simp and ring. The full problem is shown in Figure 7, which illustrates the separation between syntactic and semantic difficulty across spec, implementation, and proofs.

Notably, Claude-3.7 in combination with COPRA successfully solves every implementation certification task that any other approach is able to solve. Figure 22 in Appendix A.5 illustrates one such case, showcasing a 35-line proof for the Brazilian factorial task that requires symbolic reasoning over factorial identities and recursive structure.

Unlike math-focused benchmarks such as MiniF2F [38], where many proofs are short, goal-directed, and amenable to automation via tactics like linarith, ring, or simp, the proofs in our benchmark often mirror the *control flow* and *branching structure* of programs. As a result, standard automation is rarely sufficient. Correctness proofs frequently require reasoning case-by-case over pattern-matched inputs, recursive call structure, or multiple conditional branches. Even when the final goal involves simple arithmetic, the surrounding structure demands explicit handling of recursive unrolling, constructor cases, or fuel-based invariants. For example, proving correctness for recursive implementations like factorial products or root-finding procedures involves handling termination branches, intermediate values, and variable dependencies that make tactics like linarith or ring ineffective without significant manual decomposition. This structurally rich proof landscape contrasts with the often-flat logical forms seen in MiniF2F and underscores the need for symbolic agents like COPRA that can perform guided proof search beyond tactic chaining.

## 4 Related Work

**Formal Verification.** Formal verification encompasses a range of techniques aimed at mathematically proving the correctness of software or hardware systems with respect to a formal specification, thereby providing strong guarantees beyond traditional testing. Dafny and Verus [19, 18] utilize SMT solvers to perform verification given proper verification conditions. Interactive theorem provers

| Model | Approach | Certification | # Qed | Avg. # Lines | # Line (Min-Max) | Avg. Time (s) |
|---|---|---|---|---|---|---|
| GPT-4o | FS | Spec | 1 | 16.0 | 16–16 | 124.3 |
| GPT-4o | FS | Impl | 1 | 6.0 | 6–6 | 291.6 |
| o4-mini | FS | Spec | 2 | 29.5 | 26–33 | 87.0 |
| o4-mini | FS | Impl | 3 | 14.0 | 10–21 | 204.0 |
| Claude-3.7 | FS | Spec | 1 | 38.0 | 38–38 | 195.7 |
| Claude-3.7 | FS | Impl | 3 | 12.7 | 6–21 | 414.4 |
| DeepSeek-R1 | FS | Spec | 1 | 26.0 | 26–26 | 170.8 |
| DeepSeek-R1 | FS | Impl | 9 | 14.1 | 3–27 | 137.73 |
| GPT-4o | COPRA | Spec | 3 | 26.3 | 16–44 | 97.9 |
| GPT-4o | COPRA | Impl | 6 | 10.8 | 6-19 | 199.6 |
| Claude-3.7 | COPRA | Spec | 2 | 30.5 | 16–45 | 308.7 |
| Claude-3.7 | COPRA | Impl | 14 | 14.3 | 4–35 | 165.8 |

Table 2: Analysis of successfully generated proofs across different models and certification types. We report: (1) the number of problems for which the correctness (isomorphism resp.) proofs are found by the approach in the column "# Qed" (see Figure 1), (2) the average number of lines in the proof, (3) the range of proof lengths (min–max), and (4) the average time it took for the approach to find a proof (given a proof was found). This analysis highlights variation in proof complexity and model behavior across settings. Few-shot prompting typically yields shorter, more brittle proofs, while COPRA-augmented configurations show higher robustness, with more consistent success and a broader range of proof strategies. Proof line counts serve as a coarse indicator of reasoning complexity.

```
(a)

def problem_spec (impl : Int → Int → Int) (x y :
    Int) :=
  let spec (res : Int) := res - x - y = 0
  ∃ result, impl x y = result ∧ spec result

(b)

def generated_spec (impl : Int → Int → Int) (x y :
    Int) : Prop :=
  impl x y = x + y

(c)

def implementation (x y : Int) : Int := x + y

(d)

theorem correctness (x y : Int) : problem_spec
    implementation x y :=
by
  unfold problem_spec
  let result := implementation x y
  use result
  simp [result]
  simp [implementation]
```

```
(e)

theorem spec_isomorphism :
  ∀ impl, (∀ x y, problem_spec impl x y) ↔
          (∀ x y, generated_spec impl x y) :=
by
  intro impl
  apply Iff.intro
  -- → direction
  intro h_prob_spec
  intro x y
  have h := h_prob_spec x y
  simp [generated_spec, problem_spec] at h
  rw [generated_spec]
  linarith
  -- ← direction
  intro h_gen_spec
  intro x y
  unfold problem_spec
  simp
  have h := h_gen_spec x y
  simp [generated_spec] at h
  rw [h]
  ring
```

Figure 7: **End-to-end verified example: Problem 53 (Add Two Numbers).** This task requires adding two integers x and y. Shown are all components of the certification pipeline: (a) a non-computable ground truth spec using subtraction to hide the implementation, (b) the model-generated spec, (c) the implementation x + y, (d) a short correctness proof, and (e) an isomorphism proof relating the two specs. While the implementation is simple, the spec equivalence proof requires symbolic reasoning. This is the only HumanEval-derived task with full verification across multiple approaches.

like Lean, Isabelle, and Coq [6, 29, 11] offer highly expressive logics where users construct proofs interactively with tactic-based automation. Notably, interactive theorem provers have been involved in the verification of C compilers, microkernels, and distributed systems protocols [20, 15, 34].

**Benchmarks.** Recent efforts have developed benchmarks for formal verification with the onset of powerful neural models. FVAPPS [9] uses an LLM on scraped competition problems to automatically create formal specifications for 4715 problems, 1083 of which are guarded with test cases. However, the formal specifications themselves are often easily hackable (see Appendix A.1), with verification correctness guarded by a layer of test cases. Here, we aim to provide complete formal specifications,

which cannot be done accurately with automatic annotation. miniCodeProps [26] contains 201 verification problems regarding data structures and induction problems; however, they do not include specification synthesis or equivalence tests. DafnyBench [27] is a benchmark of 782 stand-alone Dafny programs collected from prior benchmarks and Dafny repositories, where the synthesis task is to generate the verification conditions that allow Dafny to prove correctness. At the time, the best model was Claude 3 Opus which solved $\approx 68\%$ of the problems. Software engineering benchmarks have become extremely popular in recent literature, including benchmarking performance fixing real-world issues [14] and contamination-free code generation [12]. In our work, we employ HUMANEVAL [4] to create CLEVER, our formal verification and synthesis benchmark. Formal verification is also applied in mathematical domains. Mathlib [28] and the Archive of Formal Proofs [1] constitute formal mathematical repositories in Lean and Isabelle respectively, from which benchmarks have been derived [10, 13]. ProofNet [3] serves as a benchmark for producing proper specifications of mathematical problems. PutnamBench [32] is a formal benchmark of undergraduate-level competition problems in Lean, Isabelle, and Coq.

**Proving Methods.** Recent advances in neural models and LLMs have led to increased attention on formal verification and theorem-proving. AlphaVerus [2] introduces a tree search and refinement algorithm to self-improve at producing formally verified Verus code. Similarly, SAFE [5] performs expert iteration in producing high-quality specification and proofs for generating verified Verus code. FVEL [23] uses symbolic methods to convert C programs into Isabelle, and then uses an LLM to generate correctness specifications which it then tries to prove. However, the automatic nature of the specification generation means correctness is not guaranteed. For mathematical theorem-proving, approaches involve tree search [30, 36], reinforcement learning [24, 17], LLMs [31, 35], and data augmentation and scale [8, 7].

## 5 Conclusion

We introduced a new benchmark for end-to-end verified code generation that shifts the focus from surface-level correctness to formal semantic guarantees. Unlike prior benchmarks that rely on test cases or computable specifications, our tasks are grounded in *non-computable*, logic-based specifications that are explicitly designed to prevent implementation leakage. By enforcing a separation between specification intent and implementation behavior, the benchmark demands genuine reasoning rather than pattern matching or memorization.

Our evaluation protocol is deliberately staged, decomposing the pipeline into independently checkable phases: specification generation, isomorphism proof, implementation synthesis, and correctness proof. This staged design enables fine-grained diagnosis of where models succeed and fail—whether in interpreting informal intent, aligning it with formal meaning, or synthesizing verifiably correct programs. In particular, verifying the generated specification via *isomorphism proofs* ensures semantic fidelity and introduces a novel opportunity: verified *mining* of natural language and formal specification pairs from model generations, which could be reused for bootstrapping new training data.

Our benchmark introduces challenges beyond those in mathematical theorem-proving settings like miniF2F, where proofs are often short and tactic-friendly. In contrast, our tasks reflect the branching structure of real-world programs, requiring symbolic reasoning over control flow, recursion, and invariants—scenarios where automation alone breaks down. By combining structural complexity with formal soundness, non-leakage by design, and staged verification, the benchmark offers a rigorous, semantics-grounded testbed for verified code generation. It sets a new standard for advancing neural-symbolic reasoning toward scalable, trustworthy software verification.

## Acknowledgments and Disclosure of Funding

This work was supported by NSF awards CCF-2212559 and CCF-2403211, and a 2025 Renaissance Philanthropy AI for Math award.

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

```
/--
solve_elections:
There are n voters, and two ways to convince each of them to vote for you. The
    first way to convince the i-th voter is to pay him p_i coins. The second way is
    to make m_i other voters vote for you, and the i-th voter will vote for free.
    Moreover, the process of such voting takes place in several steps. For example,
     if there are five voters with m_1 = 1, m_2 = 2, m_3 = 2, m_4 = 4, m_5 = 5, then
    you can buy the vote of the fifth voter, and eventually everyone will vote for
    you. Set of people voting for you will change as follows: 5 → 1, 5 → 1, 2, 3, 5
    → 1, 2, 3, 4, 5. Calculate the minimum number of coins you have to spend so that
    everyone votes for you.
-/

def solve_elections (n : Nat) (voters : List (Nat × Nat)) : Nat := 0

theorem solve_elections_nonnegative (n : Nat) (voters : List (Nat × Nat)) :
    solve_elections n voters >= 0 :=
by rfl

theorem solve_elections_upper_bound (n : Nat) (voters : List (Nat × Nat)) :
    solve_elections n voters <= List.foldl (λ acc (pair : Nat × Nat) => acc + pair
    .2) 0 voters :=
Nat.zero_le _

theorem solve_elections_zero_votes (n : Nat) (voters : List (Nat × Nat)) : (List.
    all voters (fun pair => pair.1 = 0)) -> solve_elections n voters = 0 :=
fun _ => rfl

theorem solve_elections_single_zero_vote : solve_elections 1 [(0, 5)] = 0 :=
by rfl
```

Figure 8: FVAPPS sample 23 and a trivial program that solves it, illustrating the limitations of not verifying full program behavior.

## A   Appendix

### A.1   FVAPPS Benchmark

The FVAPPS benchmark [9] is another code generation benchmark in Lean. However, unlike CLEVER, which requires a comprehensive proof of full program behavior, FVAPPS only requires the proof of a limited selection of properties of the program. The limitations of this are illustrated by the FVAPPS example in Figure 8. Here, a problem with a relatively complex natural language description only requires verifying lower-bound and upper-bound properties of the program implementation, as well as a few simple base cases. As can be seen, these properties are provably satisfied by a trivial program that always outputs 0 regardless of the input. Thus, it is clear that only requiring the proof of a small handful of properties does not capture the full intent of the natural language problem. This highlights the necessity of a verified code generation benchmark to require proofs of full program behavior, not just program properties.

### A.2   Hard to write Specifications

Figure 9 shows some problems for which the formal specification or the implementation is hard to write.

### A.3   Writing non-computable specifications

Figure 10 shows a computable vs non-computable version of the specification for finding the $n^{th}$ Fibonacci number. It can be observed that the computable version of the specification *leaks* the

```
(a)                                          (b)

def problem_spec                             def problem_spec
-- function signature                        -- function signature
(implementation: List Rat → Rat)             (implementation: Nat → Nat)
-- inputs                                     -- inputs
(xs: List Rat) :=                            (n: Nat) :=
-- spec                                       -- spec
let spec (result: Rat) :=                    let spec (result: Nat) :=
  let eps := (1: Rat) / 1000000;               n > 0 →
  xs.length ≥ 1 → xs.length % 2 = 0 →            (∃ i, Nat.fib i = result ∧ Nat.Prime
  ∀ poly : Polynomial Rat,                       result ∧
    poly.degree = some (xs.length - 1) →           (∃! S : Finset Nat, S.card = n - 1
    (∀ i, i ≤ xs.length - 1 → poly.coeff          ∧
     i = xs.get! i) →                              (∀ y ∈ S, (∃ k, y = Nat.fib k) ∧ y
    |poly.eval result| ≤ eps;                      < result ∧ Nat.Prime y))
-- program termination                           )
∃ result,
  implementation xs = result ∧
  spec result                                -- implementation without proof of
                                             -- termination
-- possible implementation using Newton's    def implementation (n: Nat) : Nat :=
     method                                  let rec fib_prime (n: Nat) (i: Nat) : Nat
def implementation (xs: List Rat) : Rat             :=
    :=                                         if Nat.Prime (Nat.fib i) then
let rec poly (xs: List Rat) (x: Rat) :=          if n = 1 ∨ n = 0
    xs.reverse.foldl (λ acc a => acc * x       then Nat.fib i
    + a) 0;                                     else fib_prime (n - 1) (i + 1)
let rec poly' (xs: List Rat) (x: Rat) :=     else fib_prime n (i + 1)
    (xs.drop 1).reverse.foldl (λ acc a =>     termination_by n
     acc * x + a) 0;                         decreasing_by
let rec eps := (1: Rat) / 1000000;             -- Proof of termination is open problem
let rec find_zero (xs: List Rat) (guess:       sorry
    Rat) (fuel: Nat) :=                        sorry
let eval := poly xs guess;                    fib_prime n 0
let eval' := poly' xs guess;
if eval ≤ eps ∨ fuel = 0 then (guess,
    fuel)
else
let guess' := (eval' * guess - eval) /
    eval';
find_zero xs guess' (fuel - 1);
(find_zero xs 1.0 1000000).1
```

Figure 9: Examples of benchmark challenges. (a) Polynomial root-finding: difficulties in proving termination of numerical search; (b) Prime Fibonacci finder: problem complexity rooted in the lack of a known proof of infinitude.

```
(a)                                              (b)

-- computable spec                               -- non-computable spec
def problem_spec                                 inductive fibonacci_non_computable : ℕ →
-- function signature                                   ℕ → Prop
(implementation: List Nat → Nat)                 | base0 : fibonacci_non_computable 0 0
-- inputs                                        | base1 : fibonacci_non_computable 1 1
(n: Nat) :=                                      | step  : ∀ n f₁ f₂,
-- spec                                          fibonacci_non_computable n f₁ →
let spec (result: Nat) :=                        fibonacci_non_computable (n + 1) f₂ →
  (n = 0 → result = 0) ∨                         fibonacci_non_computable (n + 2) (f₁ + f₂)
  (n = 1 → result = 1) ∨
  (2 ≤ n → ∃ fib_array : List Nat,               def problem_spec
  fib_array.length = n + 1 ∧                     -- function signature
  fib_array[0]! = 0 ∧                            (implementation: Nat → Nat)
  fib_array[1]! = 1 ∧                            -- inputs
  (∀ i, 1 < i → i < n + 1 →                      (n: Nat) :=
  fib_array[i]! = fib_array[i - 1]! +            -- spec
  fib_array[i - 2]!) ∧                           let spec (result: Nat) :=
  result = fib_array[n]!)                          fibonacci_non_computable n result
-- program termination                           -- program termination
∃ result,                                        ∃ result,
  implementation xs = result ∧                     implementation xs = result ∧
  spec result                                      spec result
```

Figure 10: Two different specs for finding the $n^{th}$ Fibonacci number. (a) shows a computable specification that *leaks* the implementation; (b) shows a non-computable specification leading to no-leakage of the implementation and enforcing the model to learn the deeper logical inference.

implementation in contrast to the non-computable version. The non-computable specification uses an **inductive** definition of a recursive function.

Writing *non-computable* specifications is a non-trivial task that requires a deep understanding of the problem. Figure 11 (problem 160) presents another complex example illustrating the difficulty of formulating such specifications. Figure 11 shows two versions of a specification for evaluating an expression given as a list of strings (["2","+","3","*","4","-","5"]). Figure 11(a) evaluates the expression and later checks if the output matches the result (not specified in the figure), which is computable. Figure 11(b) shows a non-computable version of the specification that checks if the result matches the output of evaluating the expression without leaking the implementation. One can notice that we need multiple inductive recursive definitions to ensure that the specification is clean and non-computable.

### A.4 Baseline Prompts

Snippets of the few-shot specification generator's system and example prompts are shown in Figure 12 and Figure 13. Snippets of the few-shot isomorphism prover's system and example prompts are shown in Figure 14 and Figure 15. COPRA's system prompt, used for both isomorphism and correctness, is nearly identical to the original one in the COPRA paper [31]. Snippets of COPRA's example prompt for isomorphism are shown in Figure 16.

Snippets of the few-shot implementation generator's system and example prompts are shown in Figure 17 and Figure 18. Snippets of the few-shot correctness prover's system and example prompts are shown in Figure 19 and Figure 20. Snippets of COPRA's example prompt for correctness are shown in Figure 21.

(a)

```
inductive Op where
  | add | sub | mul | floordiv
deriving Repr, DecidableEq

def parseOp : String → Option Op
  | "+" => some .add | "-" => some .sub
  | "*" => some .mul | "//" => some .
    floordiv
  | _ => none

def precedence : Op → Nat
  | .mul | .floordiv => 2
  | .add | .sub      => 1

def apply : Op → Int → Int → Int
  | .add, a, b => a + b
  | .sub, a, b => a - b
  | .mul, a, b => a * b
  | .floordiv, a, b => a / b

inductive Tok where
  | num : Int → Tok
  | op  : Op → Tok
deriving Repr

def tokenize : List String → Option (
    List Tok)
  | [] => some []
  | s :: t =>
    match parseOp s with
    | some o => (tokenize t).map (Tok.op
    o :: ·)
    | none   => s.toInt?.bind (fun n => (
    tokenize t).map (Tok.num n :: ·))

partial def evalPass (xs : List Tok) (ops
    : List Op) : List Tok :=
  match xs with
  | Tok.num a :: Tok.op o :: Tok.num b ::
    rest =>
    if o ∈ ops then evalPass (Tok.num (
    apply o a b) :: rest) ops
    else Tok.num a :: Tok.op o ::
    evalPass (Tok.num b :: rest) ops
  | x :: xs => x :: evalPass xs ops
  | [] => []

def evalTokens (tokens : List Tok) :
    Option Int :=
  let result := [[.mul, .floordiv], [.add,
    .sub]].foldl evalPass tokens
  match result with | [Tok.num n] => some
    n | _ => none

def do_algebra (input : List String) :
    Option Int :=
  tokenize input >>= evalTokens
```

(b)

```
def applyOp (x y : Int) : String →
    Option Int
  | "+"  => some (x + y)
  | "-"  => some (x - y)
  | "*"  => some (x * y)
  | "//" => if y == 0 then none else some
    (x / y)
  | _       => none

inductive evalArith_pass : List String →
    Int → Prop
| num {s : String} {n : Nat} (h : s.toNat
    ! = n) :
    evalArith_pass [s] (Int.ofNat n)
| binOp {ts1 ts2 : List String} {op :
    String} {r1 r2 r : Int}
    (h1 : evalArith_pass ts1 r1)
    (h2 : evalArith_pass ts2 r2)
    (hop : applyOp r1 r2 op = some r) :
    evalArith_pass (ts1 ++ op :: ts2) r

inductive evalArith_mul : List String →
    Int → Prop
| of_pass {ts r} (h : evalArith_pass ts r)
    : evalArith_mul ts r
| step {ts1 ts2 r1 r2 r} (h1 :
    evalArith_mul ts1 r1) (h2 :
    evalArith_mul ts2 r2)
    (hop : applyOp r1 r2 "*" = some r ∨
    applyOp r1 r2 "//" = some r) :
    evalArith_mul (ts1 ++ "*" :: ts2) r

inductive evalArith_add : List String →
    Int → Prop
| of_mul {ts r} (h : evalArith_mul ts r) :
    evalArith_add ts r
| step {ts1 ts2 r1 r2 r} (h1 :
    evalArith_add ts1 r1) (h2 :
    evalArith_add ts2 r2)
    (hop : applyOp r1 r2 "+" = some r ∨
    applyOp r1 r2 "-" = some r) :
    evalArith_add (ts1 ++ "+" :: ts2) r

-- Noncomputable spec to evaluate an
    expression
def do_algebra (input : List String) (
    result : Int) : Prop :=
  evalArith_add input result
```

Figure 11: Two different specs for evaluating an expression (as a list of strings):
["2","+","3","*","4","-","5"]. (a) shows a computable specification that evaluates using
*do_algebra*, and later checked with the result (b) shows a non-computable specification using an
inductive definition where *do_algebra* checks if the result matches the value of the expression without
leaks.

You are a good Lean 4 programmer. You are given a natural language specification of
    a function (mentioned in as a python docstring). Your task is to generate a
    Lean 4 proposition with a mentioned signature. The proposition takes in an
    implementation and program input as parameters. The proposition should hold
    true for all possible inputs in the domain, which means any preconditions
    should be mentioned in the specification to ensure that those cases are handled
     appropriately and hence the proposition is always valid if the implementation
    is correct.

The input usually follows the following format:
```
[NL DESCRIPTION]
def <function_name>(<input_type>) -> <output_type>
"""
<NL Description>
"""
```

Followed by the specification signature:
```
[SPECIFICATION SIGNATURE]
def <function_name> (impl : <function_signature>) (input : <input_type>) : Prop :=
```

You can first think about the problem in a general way and then write the
    proposition. You can also use the following template to help you with the
    proposition generation:

```
[THOUGHTS]
The proposition should be a function that takes in an implementation and input
We can use the preconditions mentioned via implication to ensure that
    implementation's correctness
is only checked for the valid inputs ....
[END THOUGHTS]
```

Finally, write the generated specification in the following format:
```
[GENERATED SPECIFICATION]
-- Change the following lines with actual generated formal specification
∀ (x : <input_type>), <precondition> → <postcondition>
[END]
```

Please closely follow the format as shown in the examples below. Make sure that
    your response always ends with [END]. Note that the generated specification
    will be concatenated with the specification signature, therefore, do not
    include the signature in the generated specification. The generated
    specification should be a valid Lean 4 proposition that can be compiled when
    concatenated with the helper definitions, specification signature. DO NOT ever
    use the `in` keyword, it is not a valid keyword in Lean 4. Please do NOT use `
    sorry` in your proof anywhere. The proof must be complete and valid.

Figure 12: Snippets of the few-shot specification generator's system prompt.

```
`example_user`
[NL DESCRIPTION]
def find_magnitude(x: int) -> int
"""
Given an integer x, your task is to find the magnitude of x.
The magnitude of an integer is defined as the absolute value of the integer.
"""

[SPECIFICATION SIGNATURE]
def generated_spec
-- function signature
(impl: Int → Int)
-- inputs
(x: Int) : Prop :=

`example_assistant`
[THOUGHTS]
We need to find absolute value of an integer.
Since absolute value is always defined for all integers, we don't need to check for
     any preconditions.
We can write a specification which return x if x is greater than or equal to 0,
     otherwise -x.
It is also easy to see that program will always terminate for all integers. However,
     it is better to mention that in the specification.
[END THOUGHTS]

[GENERATED SPECIFICATION]
∃ result, impl x = result ∧
(x >= 0 → result = x) ∧
(x < 0 → result = -x)
[END]
```

Figure 13: Snippets of the few-shot specification generator's example prompt.

```
You are a good Lean 4 programmer. You are given:
1. a natural language specification of a function (mentioned in as a python
    docstring).
2. a corresponding problem specification in lean 4.
3. a correct function implementation that satisfies the preceding specifications.

Your task is to write a formal proof in Lean 4 that the function implementation is
    correct and satisfies the formal specification.
The correctness statement is stated in the following format:
1. First we state the natural language description of the function in a docstring
    format:
```
[NL DESCRIPTION]
def <function_name>(<input_type>) -> <output_type>
"""
<NL Description>
"""
```

........

4. Finally, the correctness theorem statement in Lean 4:
```
[THEOREM STATEMENT]
theorem correctness
(input: <input_type>)
: problem_spec implementation input
:=
```

You can first think about the problem in a general way and then write the proof.
    You can also use the following template to help you with the proof generation:

```
[THOUGHTS]
The implementation should match the problem_spec in the first case because ....
The implementation should match the problem_spec in the second case because ....
....
[END THOUGHTS]
```

Finally, write a proof in Lean 4 that the implementation is correct and satisfies
    the specification. The proof should be in the following format:
```
[PROOF]
-- proof tactics
by
rw [...]
simp ...
-- more proof tactics
[END]
```

Please closely follow the format as shown in the examples below. Make sure that
    your response always ends with [END]. The proof must be in Lean 4 and should be
     valid. Make sure to always start your proof with `[PROOF]` followed by a `by`
    keyword. The proof should end with `[END]`. DO NOT forget to write the PROOF
    section. The proof section must be there in your response. Please do NOT use `
    sorry` in your proof anywhere. The proof must be complete and valid.
```

Figure 14: Snippets of the few-shot isomorphism prover's system prompt.

```
`example_user`
[NL DESCRIPTION]
def find_magnitude(x: int) -> int
""" Given an integer x, your task is to find the magnitude of x.
The magnitude of an integer is defined as the absolute value of the integer. """

[GROUND TRUTH SPECIFICATION]
def problem_spec
-- function signature
(impl: Int → Int)
-- inputs
(x: Int) :=
-- spec
let spec (result: Int) :=
(result ≥ 0) ∧
(impl (-x) = result) ∧
(result = 0 ↔ x = 0) ∧
(0 ≤ x → impl (-x) + result = 2 * x) ∧
(x ≤ 0 → impl (-x) + result = -2 * x)
-- program terminates
∃ result, impl x = result ∧
-- return value satisfies spec
spec result

[GENERATED SPECIFICATION]
def generated_spec
........

[THEOREM STATEMENT]
theorem spec_isomorphism:
∀ impl,
(∀ x, problem_spec impl x) ↔
(∀ x, generated_spec impl x) :=

`example_assistant`
[THOUGHTS]
To prove the equivalence of the two specifications, we can start by analyzing the
    generated specification.
The original specification doesn't reveal the implementation details, but we can
    see that the generated specification is a simple if-else statement that checks
    if x is less than 0.
We can use this information to write a proof that shows that the two specifications
    are equivalent.
The idea is to try different cases for x, and show that the properties of the
    generated specification hold true for the original specification as well.
[END THOUGHTS]

[PROOF]
by
unfold problem_spec
unfold generated_spec
simp
intro impl
apply Iff.intro
intro h_prob_spec
intro x
by_cases h_x_lt_0: x < 0
-- if x < 0 then
........
linarith
[END]
```

Figure 15: Snippets of the few-shot isomorphism prover's example prompt.

```
`example_user`
Goals to prove:
[GOALS]
[GOAL] 1
impl (-x) = impl x ∧
  (impl x = 0 ↔ x = 0) ∧ (0 ≤ x → impl (-x) + impl x = 2 * x) ∧ (x ≤ 0 → impl
    (-x) + impl x = -(2 * x))
[HYPOTHESES] 1
[HYPOTHESIS] impl : ℤ → ℤ
[HYPOTHESIS] h_generated_spec : ∀ (x : ℤ), impl x = if x < 0 then -x else x
[HYPOTHESIS] x : ℤ
[HYPOTHESIS] h_x_lt_0 : x < 0
[HYPOTHESIS] h_not_0_lt_x : ¬0 < x
[HYPOTHESIS] h_impl : impl x = -x
[HYPOTHESIS] h_0_le_impl_x : 0 ≤ impl x

[GOAL] 2
0 ≤ impl x ∧
  impl (-x) = impl x ∧
    (impl x = 0 ↔ x = 0) ∧ (0 ≤ x → impl (-x) + impl x = 2 * x) ∧ (x ≤ 0 →
    impl (-x) + impl x = -(2 * x))
[HYPOTHESES] 1
[HYPOTHESIS] impl : ℤ → ℤ
[HYPOTHESIS] h_generated_spec : ∀ (x : ℤ), impl x = if x < 0 then -x else x
[HYPOTHESIS] x : ℤ
[HYPOTHESIS] h_x_lt_0 : ¬x < 0

[STEPS]
[STEP] unfold problem_spec
[STEP] unfold generated_spec
[STEP] simp
[STEP] intro impl
[STEP] apply Iff.intro
[STEP] intro h_prob_spec
........
[STEP] have h_0_le_impl_x: 0 ≤ impl x := by
[STEP]    simp [h_impl]
[STEP]    linarith
[STEP] simp [h_0_le_impl_x]

[LAST STEP]
linarith [h_impl, h_0_le_impl_x, h_not_0_lt_x]

[ERROR MESSAGE]
linarith failed to find a contradiction
case pos
impl : ℤ → ℤ
h_generated_spec : ∀ (x : ℤ), impl x = if x < 0 then -x else x
x : ℤ
h_x_lt_0 : x < 0
h_not_0_lt_x : ¬0 < x
h_impl : impl x = -x
h_0_le_impl_x : 0 ≤ impl x
⊢ False
failed
[END]

`example_assistant`
[RUN TACTIC]
have h_impl_neg_x := h_generated_spec (-x)
[END]
```

Figure 16: Snippets of COPRA's example prompt for isomorphism.

```
You are a good Lean 4 programmer. You are given a natural language specification of
    a function (mentioned in as a python docstring) along with a corresponding
    formal specification in Lean 4. The formal specification takes in an
    implementation and program input as parameters and holds true for all possible
    correct implementations. Your task is to generate a Lean 4 definition with a
    mentioned signature. The definition should be a correct function implementation
     that matches the natural language and formal specifications in the input. Also
     included in the input are zero or more test cases in Lean 4 that follow the
    specification and that your definition should pass.

The input usually follows the following format:
1. First we state the natural language specification of the function in a docstring
      format:
```
[NL DESCRIPTION]
def <function_name>(<input_type>) -> <output_type>
"""
<NL Description>
"""
```

2. Followed by the formal specification in Lean 4:
........

4. Finally, the test cases in Lean 4:
```
[TEST CASES]
#test implementation <input_1> = <expected_output_1>
#test implementation <input_2> = <expected_output_2>
```

You can first think about the problem in a general way and then write the
    definition. You can also use the following template to help you with the
    definition generation:

```
[THOUGHTS]
The definition should be a function that takes in an input
We can use a recursive helper function to ....
[END THOUGHTS]
```

Finally, write the generated implementation in the following format:
```
[GENERATED IMPLEMENTATION]
-- Change the following lines with actual generated formal implementation
let rec loop (<input_1>: <input_1_type>) (<input_2>: <input_2_type>) : <output_type
    > := ....
[END]
```

Please closely follow the format as shown in the examples below. Make sure that
    your response always ends with [END]. Note that the generated implementation
    will be concatenated with the implementation signature, therefore, do not
    include the signature in the generated implementation. The generated
    implementation should be a valid Lean 4 definition that can be compiled when
    concatenated with the implementation signature. Try to write an implementation
    where termination can be automatically verified; for example, always use
    library functions (`Int.lcm`, `String.find`, `Nat.fermatNumber`, etc.) or `
    match` statements if possible. DO NOT ever use the `in` keyword, it is not a
    valid keyword in Lean 4.
```

Figure 17: Snippets of the few-shot implementation generator's system prompt.

```
`example_user`
[NL DESCRIPTION]
def find_magnitude(x: int) -> int
"""
Given an integer x, your task is to find the magnitude of x.
The magnitude of an integer is defined as the absolute value of the integer.
"""

[SPECIFICATION]
def problem_spec
-- function signature
(impl: Int → Int)
-- inputs
(x: Int) :=
-- spec
let spec (result: Int) :=
(result ≥ 0) ∧
(impl (-x) = result) ∧
(result = 0 ↔ x = 0) ∧
(0 ≤ x → impl (-x) + result = 2 * x) ∧
(x ≤ 0 → impl (-x) + result = -2 * x)
-- program terminates
∃ result, impl x = result ∧
-- return value satisfies spec
spec result

[IMPLEMENTATION SIGNATURE]
def implementation (x: Int) : Int :=

[TEST CASES]
#test implementation (-1) = 1
#test implementation 0 = 0
#test implementation 5 = 5
#test implementation (-100) = 100
#test implementation 100 = 100

`example_assistant`
[THOUGHTS]
If `x` is less than 0 then return `-x`, otherwise return `x`.
[END THOUGHTS]

[GENERATED IMPLEMENTATION]
if x < 0 then -x else x
[END]
```

Figure 18: Snippets of the few-shot implementation generator's example prompt.

```
You are a good Lean 4 programmer. You are given:
1. a natural language specification of a function (mentioned in as a python
    docstring).
2. a corresponding problem specification in lean 4.
3. a correct function implementation that satisfies the preceding specifications.

Your task is to write a formal proof in Lean 4 that the function implementation is
    correct and satisfies the formal specification.
The correctness statement is stated in the following format:
1. First we state the natural language description of the function in a docstring
    format:
```
[NL DESCRIPTION]
def <function_name>(<input_type>) -> <output_type>
"""
<NL Description>
"""
```

2. Followed by the formal specification in Lean 4:
........

4. Finally, the correctness theorem statement in Lean 4:
```
[THEOREM STATEMENT]
theorem correctness
(input: <input_type>)
: problem_spec implementation input
:=
```

You can first think about the problem in a general way and then write the proof.
    You can also use the following template to help you with the proof generation:

```
[THOUGHTS]
The implementation should match the problem_spec in the first case because ....
The implementation should match the problem_spec in the second case because ....
....
[END THOUGHTS]
```

Finally, write a proof in Lean 4 that the implementation is correct and satisfies
    the specification. The proof should be in the following format:
```
[PROOF]
-- proof tactics
by
rw [...]
simp ...
-- more proof tactics
[END]
```

Please closely follow the format as shown in the examples below. Make sure that
    your response always ends with [END]. The proof must be in Lean 4 and should be
     valid. Make sure to always start your proof with `[PROOF]` followed by a `by`
    keyword. The proof should end with `[END]`. DO NOT forget to write the PROOF
    section. The proof section must be there in your response. Please do NOT use `
    sorry` in your proof anywhere. The proof must be complete and valid.
```

Figure 19: Snippets of the few-shot correctness prover's system prompt.

```
`example_user`
[NL DESCRIPTION]
def find_magnitude(x: int) -> int
"""
Given an integer x, your task is to find the magnitude of x.
The magnitude of an integer is defined as the absolute value of the integer.
"""

[SPECIFICATION]
def problem_spec
........

[IMPLEMENTATION]
def implementation (x: Int) : Int :=
if x < 0 then -x else x

[THEOREM STATEMENT]
theorem correctness
(x: Int)
: problem_spec implementation x :=

`example_assistant`
[THOUGHTS]
We can start by unfolding the `problem_spec` and assigning the implementation's
    output to a temporary variable `result`.
Early on, we will want to break the proof up into cases based on whether `x` is
    positive, negative, or zero.
Many of the intermediate goals can likely be proven using `linarith`.
[END THOUGHTS]

[PROOF]
by
unfold problem_spec
let result := implementation x
use result
simp [result]
simp [implementation]
by_cases h_0_eq_x: x = 0
-- if x = 0 then
simp [h_0_eq_x]
by_cases h_x_lt_0: x < 0
-- if x < 0 then
simp [h_x_lt_0]
apply And.intro
linarith
apply And.intro
intro; linarith
........
by_cases h_0_lt_x: 0 < x
-- if 0 < x then
simp [h_0_lt_x]
linarith
have h_x_eq_0: x = 0 := by
  linarith
simp [h_x_eq_0]
intro h_x_le_0
simp at h_x_lt_0
have h_x_eq_0: x = 0 := by
  linarith
contradiction
[END]
```

Figure 20: Snippets of the few-shot correctness prover's example prompt.

```
`example_user`
Goals to prove:
[GOALS]
[GOAL] 1
(0 ≤ if x < 0 then -x else x) ∧
  ((if 0 < x then x else -x) = if x < 0 then -x else x) ∧
    ((if x < 0 then -x else x) = 0 ↔ x = 0) ∧
      (0 ≤ x → ((if 0 < x then x else -x) + if x < 0 then -x else x) = 2 * x) ∧
        (x ≤ 0 → ((if 0 < x then x else -x) + if x < 0 then -x else x) = -(2 * x))
[HYPOTHESES] 1
[HYPOTHESIS] x : ℤ
[HYPOTHESIS] result : ℤ := implementation x

[STEPS]
[STEP] unfold problem_spec
[STEP] let result := implementation x
[STEP] use result
[STEP] simp [result]
[STEP] simp [implementation]

[INCORRECT STEPS]
[STEP] cases x with n

[LAST STEP]
cases lt_or_ge x 0 with hx hx

[ERROR MESSAGE]
unknown tactic
[END]

`example_assistant`
[RUN TACTIC]
by_cases h_0_eq_x: x = 0
[END]
```

Figure 21: Snippets of COPRA's example prompt for correctness.

## A.5  Some Proof Found

Figure 22 shows an example of a proof found for implementation certification by Claude-3.7 using COPRA.

```
(a)
def problem_spec (impl : Nat → Nat) (n : Nat) :=
  let spec (result : Nat) :=
    let factorial := Nat.factorial n;
    (0 < n → result / factorial = impl (n - 1)) ∧
    (n = 0 → result = 1);
  ∃ result, impl n = result ∧ spec result

(b)
def implementation (n : Nat) : Nat :=
  match n with
  | 0 => 1
  | n+1 => Nat.factorial (n+1) * implementation n

(c)
#test implementation 4 = 288

(d)
theorem correctness (n : Nat) : problem_spec implementation n := by
  unfold problem_spec
  let result := implementation n
  use result
  simp [result]
  apply And.intro
  unfold implementation
  simp [Nat.factorial_succ]
  intro h_n_pos
  cases n
  exfalso
  exact h_n_pos.false
  rename_i n
  simp [Nat.succ_eq_add_one, Nat.add_sub_cancel]
  rw [Nat.factorial_succ]
  have h_fac_pos : 0 < (n + 1) * n.factorial := by
    exact Nat.mul_pos (Nat.succ_pos n) (Nat.factorial_pos n)
  have h_cancel : (n + 1) * n.factorial * implementation n
    / ((n + 1) * n.factorial) = implementation n := by
    rw [Nat.mul_div_cancel_left (implementation n) h_fac_pos]
  simp [h_cancel]
  unfold implementation
  cases n
  simp [Nat.factorial_zero]
  rename_i n
  simp [Nat.add_zero]
  simp [Nat.factorial_succ]
  left
  rw [implementation.eq_def]
  simp [Nat.mul_assoc]
  cases n
  simp [Nat.zero_eq]
  rename_i n
  simp [Nat.factorial_succ]
  rw [Nat.mul_assoc]
  intro h_n_eq_0
  rw [h_n_eq_0, implementation]
```

Figure 22: **Problem 139 (Brazilian Factorial)**: Given an integer n, compute the product of all factorials from n! down to 1!. Part (a) defines the **ground truth specification**, which expresses recursive structure without leaking the implementation. Part (b) shows the **implementation** using a recursive product of factorials. Part (c) lists a **test case** used for validation. Part (d) presents the full **correctness proof**, showing that the implementation satisfies the spec. This proof, generated by COPRA using Claude-3.7, spans 35 lines and involves reasoning over factorial identities, case analysis, and symbolic manipulation.

