# OpenReview forum: "CLEVER: A Curated Benchmark for Formally Verified Code Generation"
_NeurIPS.cc/2025/Datasets_and_Benchmarks_Track — NeurIPS 2025 Datasets and Benchmarks Track poster_

### Official Review · Reviewer_j4Pt · 2025-07-02

**Rating:** 5
**Confidence:** 4

**Summary:**

This paper introduces CLEVER, a benchmark dataset designed to evaluate LLMs on formally verified code generation. It consists of 161 carefully crafted Lean specifications derived from programming problems in the existing HumanEval dataset. CLEVER defines two interconnected tasks: specification generation and Lean implementation generation. For evaluation, the authors assess existing API-based models using few-shot and agentic approaches. Experimental results reveal consistently poor performance across models, establishing CLEVER as a challenging benchmark at the frontier of formal code generation.

**Dataset Code Accessibility:**

Yes

**Dataset Code Comments:**

- The dataset is publicly available and comes with clear instructions for running the tasks, supporting reproducibility and ease of adoption.

**Ethical Considerations:**

No, there are no or only very minor ethics concerns

**Final Justification:**

I appreciate the authors’ responses. Incorporating more advanced neural theorem provers would strengthen the experimental evaluation. I have also reviewed the other comments and remain positive about the paper.

**Limitations Weaknesses:**

I do not see any major weaknesses in this paper for a DB track, as the proposed benchmark and tasks are highly valuable and well-aligned with the track’s goals. One potential improvement would be to evaluate existing neural theorem provers, such as DeepSeek Prover and Kimina Prover, to better position the benchmark in the context of current state-of-the-art models.

**Strengths Contributions:**

- The paper is well-written, well-organized, and clearly motivated.
- The proposed dataset is carefully curated, requiring significant manual effort and domain expertise. The tasks are thoughtfully designed to prevent models from exploiting shortcuts that are present in existing datasets.
- The experimental results convincingly demonstrate the difficulty of the dataset, leaving ample room for future research and improvements.

---

> ### Author Rebuttal · Authors · 2025-07-31
>
> We thank the reviewer for their thoughtful, constructive, and positive assessments of CLEVER. We appreciate the careful consideration of our work.
>
> Thank you for your suggestion! Investigating the latest LLM approaches for theorem proving on Clever is crucial to better understand where our benchmark stands in the current research landscape. We’ll be sure to test out DeepSeekProverV2 and Kimina-Prover (the latest version from July 10) on CLEVER and include the results in the final version. We are also keeping an eye on IMO gold-worthy (such as ByteDance’s SeedProver, Harmonic’s Aristotle) models being open-sourced or available through API for evaluating as well. Additionally, we plan to maintain a leaderboard for CLEVER and will continue adding new results as they are reported by other authors on our benchmark.

---

> > ### Comment · Reviewer_j4Pt · 2025-08-05
> >
> > Thank you for the response. I’ve also read the comments from other reviewers and still maintain a positive overall impression of the paper. I will keep my score unchanged.

---

### Official Review · Reviewer_FjJk · 2025-07-03

**Rating:** 5
**Confidence:** 3

**Summary:**

Interactive theorem provers are important in formal program verifications. To test the ability of LLMs on code generation, an important criteria is to verify the generated codes with specifications. This benchmark achieve this with two tasks: first, it let LLMs to generate a computable specification that is equivalent to a given non-computable specification; then, the LLMs are prompted to generate the correctness proof. Both tasks can be exactly verified using Lean's type checkers. This approach prevents LLMs from copying intended semantics and test their real abilities of generating verified codes.

The benchmark includes 161 problems by adapting the HUMANEVAL dataset. In the experiments, it is shown that state-of-the-art open and close LLMs are all struggling with most of the problems in the benchmark, despite that they are relatively easy for humans to understand and solve. Within 600 seconds, only 1 out of 161 problems are completely solved by any of the existing LLMs. Also, it is discovered that with COPRA augmentation, all LLMs could work better at generating specifications.

**Dataset Code Accessibility:**

Yes

**Dataset Code Comments:**

The instructions and codes to test with the benchmark are available on GitHub.

**Ethical Considerations:**

No, there are no or only very minor ethics concerns

**Final Justification:**

I appreciate the authors for the detailed responses. My questions were mainly about how existing LLMs can improve their performances on this benchmark during test time. I believe there are a lot more directions to explore in the future and they are all exciting.

I have also read the other reviews and remain positive about the paper.

**Limitations Weaknesses:**

I don't spot major weaknesses but have some questions in mind.

- I am curious of the number of secs $k=600$. Why specifically choose this time period in the testing? Will the LLMs perform better after increasing $k$?

- I notice that LLMs do not always generate compilable codes in this benchmark. I am wondering if introducing constraint generation is helpful in improving the abilities of LLMs to solve more problems.

**Strengths Contributions:**

- This benchmark involves a significant amount of human efforts to write non-computable specifications, which avoid data leakage even on simple problems, leading to the challenging task. In addition, the evaluation of the benchmark involves reliable tools like Lean's type checker, making the whole pipeline rigorous. I think this is a role model for how LLM benchmarks should be created.
- The idea of letting LLMs generate computable specifications that matches non-computable specifications is very interesting. It also discusses challenges of performing equivalence proof with approximate answers. I think the investigation of this idea is thorough.

---

> ### Author Rebuttal · Authors · 2025-07-31
>
> We thank the reviewer for their thoughtful, constructive, and positive assessments of CLEVER. We are grateful for their careful consideration of our work. We would also like to address some of the questions asked by the reviewer:
>
> > I am curious of the number of secs $k=600$. Why specifically choose this time period in the testing? Will the LLMs perform better after increasing $k$?
>
> **Response**: The 600-second limit was chosen to balance computational cost with evaluation depth. We agree that increasing the time limit would likely improve performance, and this parameter can be adjusted based on evaluation needs. In [1, 2] it was observed that increasing the timeout increased the number of problems solved. We selected 600 seconds as a reasonable baseline that captures meaningful differences between approaches while remaining computationally feasible.
>
> > I notice that LLMs do not always generate compilable codes in this benchmark. I am wondering if introducing constraint generation helps improve the abilities of LLMs to solve more problems.
>
> **Response**: We agree that constrained generation could help improve LLM compilation rates. This represents an interesting direction for future work that could complement our benchmark by providing better scaffolding for code generation before the formal verification step. As evident from our results, LLMs perform better at generating compilable Lean implementations than writing correct Lean proofs. Thus, we believe that LLMs will eventually improve at Lean syntax; however, performing more involved semantic reasoning, such as writing proofs, will remain a much harder challenge.
>
> [1] An In-Context Learning Agent for Formal Theorem-Proving. Thakur et. al, 2024
>
> [2] HyperTree Proof Search for Neural Theorem Proving. Lample et. al, 2022

---

> > ### Comment · Reviewer_FjJk · 2025-08-05
> > **Keeping my score**
> >
> > As this is mandatory, I copy my final justification here:
> >
> > I appreciate the authors for the detailed responses. My questions were mainly about how existing LLMs can improve their performances on this benchmark during test time. I believe there are a lot more directions to explore in the future and they are all exciting.
> >
> > I have also read the other reviews and remain positive about the paper.

---

> ### Comment · Area_Chair_2TuQ · 2025-08-03
>
> Dear reviewer,
>
> Please read the rebuttal and provide your *final justification* and score.
>
> Best,
>
> AC

---

### Official Review · Reviewer_K4aK · 2025-07-04

**Rating:** 5
**Confidence:** 3

**Summary:**

The paper introduces a new benchmark for code, specification, and proof generation. For each HumanEval problem, the authors manually create a non-computable specification that prevents LLMs from copying code or generating trivial proofs. They then evaluate LLMs on their ability to generate specifications, code, and corresponding proofs, and find that current LLMs struggle with proof generation.

**Dataset Code Accessibility:**

Yes

**Dataset Code Comments:**

The repository contains detailed instructions for using the code and data.

**Ethical Considerations:**

No, there are no or only very minor ethics concerns

**Final Justification:**

I have read the authors' response and other reviewers' comments. I have no further concern.

**Limitations Weaknesses:**

The paper could include a comparison between pass@k-seconds and pass@k-attempts using specific LLMs to present the results more clearly and enable direct comparison with prior work.

**Strengths Contributions:**

- The paper highlights a problem where specifications can leak key information, allowing LLMs to cheat when generating code and proofs. To address this, the authors use hand-crafted non-computable specifications that hide implementation details from the LLMs.

- The paper introduces a new pass@k-seconds metric to evaluate LLMs, which improves upon the traditional pass@k-attempts by accounting for computational cost. This metric balances evaluation by considering that some LLMs generate more tokens per attempt.

---

> ### Author Rebuttal · Authors · 2025-07-31
>
> We thank the reviewer for their thoughtful, constructive, and positive assessments of CLEVER. We appreciate the careful consideration of our work. We would also like to address some of the limitations pointed out by the reviewer:
>
> > The paper could include a comparison between pass@k-seconds and pass@k-attempts using specific LLMs to present the results more clearly and enable direct comparison with prior work.
>
> **Response**: We appreciate this suggestion. We chose `pass@k-seconds` specifically to enable fair comparisons between different proof generation approaches, particularly given the significant differences between agentic methods that generate proofs step-by-step versus methods that generate complete proofs in one go. We will include a detailed comparison between `pass@k-seconds` and `pass@k-attempts` across different LLMs in the final version to highlight how our metric might better account for these computational cost differences.

---

### Official Review · Reviewer_L9Pf · 2025-07-07

**Rating:** 4
**Confidence:** 3

**Summary:**

CLEVER introduces a rigorously curated benchmark that pushes code-generation systems beyond unit-test correctness toward fully machine-verified semantics. Built from 161 HumanEval-inspired problems re-expressed in Lean 4, the benchmark requires models to translate natural-language descriptions into non-computable Lean specifications and formally prove them equivalent to hidden ground-truth specs, and synthesize Lean implementations that are subsequently proved to satisfy those specs. Experiments with leading LLMs and the neuro-symbolic COPRA agent show that even state-of-the-art approaches can certify at most one task end-to-end, underscoring the difficulty of the setting. All tasks, proofs, and evaluation tools are openly released on GitHub and HuggingFace, providing the community with a challenging, reproducible test bed for future research in formally verified code generation.

**Dataset Code Accessibility:**

Yes

**Ethical Considerations:**

No, there are no or only very minor ethics concerns

**Final Justification:**

THX for your rebuttal. I will keep my positive rating.

**Limitations Weaknesses:**

1. Insights may not transfer to Coq, Isabelle, or mainstream languages; tooling barrier for non-Lean users.
2. All problems derived from HumanEval (Python-centric); lacks systems-level, parallel, or floating-point heavy tasks.
3. Extending or correcting tasks requires expert Lean authors (~40 min/problem); limits scalability/maintenance.

**Strengths Contributions:**

1. Prevents trivial code-copying; enforces genuine reasoning.
2. Shows current SOTA LLMs compile many specs/impls but fail at proofs
3. All outputs are verified by Lean’s kernel—no reliance on test cases

---

> ### Author Rebuttal · Authors · 2025-07-31
>
> Thank you so much for your positive assessments and constructive feedback on CLEVER. We would like to address some of the limitations/concerns that you have raised:
>
> > Insights may not transfer to Coq, Isabelle, or mainstream languages; tooling barrier for non-Lean users.
>
> **Response:** We appreciate this concern. Lean 4 has significant traction with Mathlib4, containing a huge number of theorems and strong support from the Lean community. Recent tactics introduced, like `grind`, will bridge some of the proof automation gaps with other languages like Coq and Isabelle.
>
> The core challenge of translating natural language to formal specs and proving correctness transfers across proof assistants, with recent efforts like CSLib in Lean 4, the transferability aspect will be circumvented due to support for multi-lingual transpilation. Moreover, CLEVER establishes a rigorous methodology that can be easily adapted to other systems.
>
>
> > All problems derived from HumanEval (Python-centric); lacks systems-level, parallel, or floating-point heavy tasks.
>
> **Response**: We agree this is a current limitation. CLEVER serves as a starting point that allows direct comparison with existing code generation benchmarks while covering diverse algorithmic challenges. We believe that harder real-world problems will be even more challenging for the AI systems, so this serves as an incrementally challenging benchmark to improve the AI systems in end-to-end verified code generation. As you can already see, our benchmark with just Human eval problems itself is quite challenging for end-to-end verified code generation. Also, our open-source release makes it easy for the community to extend to systems-level and domain-specific problems.
>
>
> > Extending or correcting tasks requires expert Lean authors (~40 min/problem); limits scalability/maintenance.
>
> **Response**: The 40 minutes per problem ensures mathematical soundness and prevents AI from cheating by producing trivial solutions, something unavoidable in existing benchmarks like FVAPPS (see Figure 7 in the Appendix). If problems were easy to generate automatically, they would be too simple to evaluate formal reasoning capabilities. The near-zero success rates from current SOTA models validate that this is a genuine test of AI abilities.

---

> ### Comment · Area_Chair_2TuQ · 2025-08-03
>
> Dear reviewer,
>
> Please read the rebuttal and provide your *final justification* and score.
>
> Best,
>
> AC

---

### Note · Authors · 2025-08-16

Thank you to the reviewers for their thoughtful comments. The feedback was very useful.

Reviewers recognized that CLEVER checks specs and proofs with Lean and that current models still fail on most tasks. We appreciate that the effort to build the benchmark was noticed by the reviewers. The suggestions to try newer provers, to expand beyond HumanEval-style problems, and to test constraint-based generation are all valuable, and we plan to explore them in future work.

Our main goal with CLEVER was to move past unit tests and see if models can generate verified specs, code, and proofs. We hope CLEVER can serve as a starting point for future research on verified code generation.

Thanks again for the constructive feedback.

---

### Decision · Program_Chairs · 2025-09-18

**Decision:**

Accept (poster)

**Comment:**

This paper introduces a well curated benchmark, CLEVER, that pushes code-generation systems beyond unit-test correctness toward fully machine-verified semantics. It is built based on 161 HumanEval-inspired problems re-expressed in Lean 4, the benchmark requires models to translate natural-language descriptions into non-computable Lean specifications and formally prove them equivalent to hidden ground-truth specs, and synthesize Lean implementations that are subsequently proved to satisfy those specs.

Experiments with leading LLMs and the neuro-symbolic COPRA agent show that even state-of-the-art approaches can certify at most one task end-to-end, underscoring the difficulty of the setting. All tasks, proofs, and evaluation tools are openly released on GitHub and HuggingFace, providing the community with a challenging, reproducible test bed for future research in formally verified code generation.

After the rebuttal, all reviewers are positive about the paper. The following limitations are still reasonable and are partially solved:
- Tool barrier for non-Lean users;
- All problems derived from HumanEval;
- Tasks requires expert Lean authors.

Further exploration is encouraged.